# Rheology of Recycled PET

**DOI:** 10.3390/ma16093358

**Published:** 2023-04-25

**Authors:** Ilaria Cusano, Laura Campagnolo, Marco Aurilia, Salvatore Costanzo, Nino Grizzuti

**Affiliations:** 1Department of Chemical, Materials and Production Engineering, University of Naples Federico II, P.le Tecchio 80, 80125 Naples, Italy; ilaria.cusano@unina.it (I.C.); salvatore.costanzo@unina.it (S.C.); 2Gurit, 10088 Volpiano, Italy; laura.campagnolo@gurit.com (L.C.); marco.aurilia@gurit.com (M.A.)

**Keywords:** polyethylene terephthalate, reactive extrusion, rheology

## Abstract

Polyethylene terephthalate (PET) is a thermoplastic material that is widely used in many application fields, such as packaging, construction and household products. Due to the relevant contribution of PET to global yearly solid waste, the recycling of such material has become an important issue. Disposed PET does not maintain the mechanical properties of virgin material, as exposure to water and other substances can cause multiple chain scissions, with subsequent degradation of the viscoelastic properties. For this reason, chain extension is needed to improve the final properties of the recycled product. Chain extension is generally performed through reactive extrusion. As the latter involves structural modification and flow of PET molecules, rheology is a relevant asset for understanding the process and tailoring the mechanical properties of the final products. This paper briefly reviews relevant rheological studies associated with the recycling of polyethylene terephthalate through the reactive extrusion process.

## 1. Introduction

Polyethylene terephthalate (PET) is the most commonly used polymer in the polyester family. Its worldwide annual production capacity reached 30.5 million metric tons in 2019, and it is expected to increase to 35.3 million metric tons by 2024 [1]. Due to its excellent barrier properties for water, moisture and oxygen, the main application of PET is packaging. Other uses include plastic bottles, fibers, construction materials, and disposable houseware. Advanced applications are concerned with surface modifications of PET to obtain rapid bactericidal and biocompatible fabrics [2] or conductive fibers by coating with carbon nanotubes [3]. Because of their extensive use, PET products are frequently dispersed in the environment, and owing to their large production volumes and non-biodegradability, relevant issues arise. PET is one of the main components of microplastics in seawater [4,5], and it was recently found to be a contaminant in human blood [6].

Given the above scenario, recycling of PET has become of primary importance. To this end, both chemical and mechanical recycling are available [7]. Chemical recycling involves the enzymatic or catalytic depolymerization of PET chains into monomers, which can be reused for processing [8]. Mechanical recycling is performed through the recovery and remolding of PET waste into new products. This route encompasses different steps such as sorting, washing and grinding of PET waste. During this cycle, PET undergoes degradation, with subsequent reduction in the molecular weight and melt strength. For this reason, mechanical recycling is generally followed by post-polymerization processes.

An efficient strategy to recover the mechanical properties of PET is chain extension. In such a process, PET is reacted with small functional molecules referred to as *chain extenders*, which can link two or more chains of PET to form either linear or branched structures. The degree of molecular branching depends on process conditions such as the concentration and functionality of the chain extender.

Due to its peculiar sensitivity to the details of the molecular architecture (e.g., molecular weight and its distribution, linear vs. branching structure), rheology is a powerful technique to detect structural changes and to optimize the final properties of the extruded products. The scientific literature encompasses several reviews about the general aspects of PET recycling [9] and chain extension of polyesters [10,11,12]. However, there is a lack of studies specifically focusing on the link between rheology and reactive extrusion. The aim of this review is to provide insight into the rheological aspects related to the recycling of PET. The review is structured as follows. First, we provide a general overview of the chemical properties of the material, the chain extension process and the most frequently used chain extenders. Then, we describe the rheological methods used to characterize PET rheology. Subsequently, we review the main results reported in rheological studies about the stability of PET, the linear and nonlinear rheological properties of virgin and reacted PET and, finally, the modeling attempts that have been proposed to relate the molecular structure of PET to its rheological properties.

## 2. Chemical and Physical Properties of PET

PET production starts with the reaction of either terephthalic acid or dimethyl phthalate with ethylene glycol to produce bis(hydroxyethyl) terephthalate. The latter undergoes a prepolymerization and polycondensation process to reach a weight-averaged molecular weight ($M_{w}$) of around $20\times 10^{3}$–$30\times 10^{3}$ g/mol. The molecular weight can be increased by solid-state polymerization up to values of the order $10^{5}$ g/mol [9]. As the reaction involves polycondensation with high degree of conversion, the polydispersity index is generally around 2 [13]. The repeating unit of PET is reported in Figure 1.

PET is a semicrystalline polymer. The degree of crystallinity depends on several parameters, such as the crystallization temperature, the molecular weight and the molecular weight distribution. A higher molecular weight induces a lower degree of crystallinity and more defects in the crystals [14]. An increase in the degree of crystallinity, on the other hand, increases the yield stress and decreases the maximum elongation at break [15].

PET can solubilize different gases, for example argon, nitrogen, and CO_2_ [16], making it a good candidate for foaming processes, which are generally coupled with reactive extrusion to improve foamability [17,18].

A comprehensive report of the chemical and physical properties of PET can be found in Table 2 of reference [9]. Here, some chemical physical properties relevant to rheology are reported in Table 1. Rheological measurements are limited by crystallization at low temperatures and by polymer degradation at high temperatures. Hence, a typical temperature range for measurements is between 260 and 320 °C. Above the latter temperature, the stability of the material rapidly decreases over time. The Flory characteristic ratio, an index of the local stiffness of the polymeric chain, is relatively low for PET. The subsequent high flexibility of PET results in a reduced entanglement molecular weight compared to stiffer molecules such as polystyrene. As a consequence, it is easy to achieve well-entangled chains (≈15 entanglements per chain), even with an $M_{w}$ of the order of 30,000 g/mol. However, since the extrusion process is carried out at high temperatures, the zero shear viscosity of virgin PET remains relatively low (around 1000 Pa s) [19].

## 3. Chain Extension

Chain extension is the process by which low-viscosity PET is reacted in order to increase its molecular weight and create branched structures. This generally occurs in a reactive extrusion process. During reactive extrusion, molten PET is mixed and reacted with small functional molecules—so-called chain extenders. Depending on the functionality and concentration of chain extenders, different structures are possible. Figure 2 depicts possible structures that can be formed through reaction with difunctional, trifunctional or tetrafunctional chain extenders. When the functionality is greater than two, dendritic structures are generated. If the extension reaction proceeds further, then crosslinking occurs, with the formation of an insoluble gel.

Chain extension can occur either by reaction of functional groups or by free-radical reactions. Anhydrides, epoxy-based chain extenders, oxazolines, imides and phosphites promote chain extension through the first mechanism, whereas reaction with peroxides involves reactive radicals [12]. For this reason, the reaction with functional groups has the advantage of enabling more control relative o that with peroxides.

A comprehensive review of the different types of chain extenders used to improve the mechanical properties of recycled PET was recently published by Jang and coworkers [10]. Here, we limit our study to the most used chain extenders; their referenced articles are listed in Table 2.

### 3.1. Pyromellitic Dianhydride

Pyromellitic dianhydride (PMDA) is currently one of the best-performing reagents in chain extension reactions [9]. PMDA is a tetrafunctional compound: the two anhydrides in its molecular structure are able to interact with the hydroxyl end groups of the PET linear chains, leading to the formation of stars or hyperbranched structures of up to four arms [33]. PMDA has the advantages of being thermally stable and not forming byproducts [34,35]. The disadvantage of PMDA is that it has low reactivity with the hydroxyl group [36]. The latter issue can be circumvented by using PMDA with other chain extenders [37]. Another disadvantage is the tendency of PMDA to absorb moisture.

### 3.2. Joncryl

Joncryl is a multifunctional epoxide oligomer that is widely used in industry [38]. During chain extension, the reaction of the epoxy group with the carboxyl end group of the PET linear chain is favored; nevertheless, interaction with the terminal hydroxyl group also takes place. Since its average value of functionality is 9, Joncryl can form star architectures with a maximum of 9 arms with PET [39]. Joncryl has the advantage of being highly reactive due to the multiple epoxy groups [12] and the ability to easily form branched structures. However, its high reactivity can also result in rapid crosslinking.

### 3.3. Tetrahydrophthalic Acid Diglycidyl Ester

Tetrahydrophthalic acid diglycidyl ester (TADE) is mainly used to increase the molecular weight of linear chains. The epoxide group of TADE mainly reacts with the carboxyl end group of the PET chain, first through an epoxide ring-opening reaction and then via the formation of covalent bonds [40]. The main drawback of TADE is that it mainly produces linear structures. Therefore, it is generally combined with PMDA to obtain branching [40].

### 3.4. Triglycidyl Isocyanurate (TGIC)

The glycidyl group of TGIC prefers to interact with the carboxyl end group of linear PET chains [41]. One of the possible branching architectures that can be formed is a star structure, with a maximum of three arms [33]. Compared to other chain extenders, the low functionality of TGIC results in a lower degree of branching.

### 3.5. Tetraglycidyl Diamino Diphenyl Methane

Like TGIC, tetraglycidyl diamino diphenyl methane (TGDDM) is among the most frequently cited epoxy-branching additives in the literature [18]. TGDDM, a tetrafunctional chain extender, also reacts predominantly with the carboxyl terminal group of PET; this leads to the formation of star-like structures, which randomly interact with each other to create more complex architectures [42]. As for all the other epoxy-based chain extenders, hydroxyl byproducts are also formed, which can react with the carboxyl or epoxy groups to form branched structures [43].

**Table 2 materials-16-03358-t002:** Chain extenders in the literature.

Chain Extender	References
PMDA	[17,18,19,33,38,39,40,42,44,45,46,47,48,49,50,51,52,53,54]
Joncryl	[29,38,39,50,52,53,54,55,56,57,58]
TADE	[40]
TGIC	[18,33,41,59]
TGDDM	[18,42,60,61,62]

The chain extension process results not only in a change in the average molecular weight distribution of the recycled PET but also in a substantial modification of the chain architecture. The latter has a strong impact on the rheological properties, especially in nonlinear extension. When the functionality of the chain extender is two, longer linear chains are formed. This has a considerable effect on the melt viscosity, which increases as $\eta_{0}\propto M_{w}^{3.4}$, where $M_{w}$ is the average molecular weight [63]. However, since the architecture is linear, the extensional properties remain scarce [13]. Conversely, by using chain extenders with higher functionality than two, branched structures are formed. The branching strongly impacts the extensional behavior, inducing strong strain hardening, which is relevant for foaming applications [12].

## 4. Rheological Methods for Testing PET Materials

Different rheological approaches were used to characterize either virgin or reacted PET samples. The most common techniques include shear rotational rheometry, extensional rheometry and capillary rheometry. This section reports a brief description of the experimental rheological techniques. A summary of how these techniques are used to measure the rheology of PET in the literature is reported in Table 3.

### 4.1. Viscometry

The molecular weight of polymers can be determined according to intrinsic viscosity [η] measurements through the well-known Mark–Houwink–Sakurada relation:(1)$$\left[\eta \right]=KM^{\alpha }$$
where *K* and α depend on the specific polymer/solvent system [13]. for example, $K=6.56\times 10^{-4}$ and α = 0.76 for PET in o-chlorophenol [64].

The Mark–Houwink relation works for monodisperse polymers and requires multiple measurements at different polymer concentrations (*C*) to determine [η]. However, empirical single-point correlations were developed to obtain the [η] of PET [65].

The intrinsic viscosity is obtained by measuring the viscosity of polymer solutions in a dilute regime. To this end, either Ubbelohde-type viscometers [38,61,64] or automated falling-ball (or rolling-ball) viscometers [66] were used. The first apparatus consists of a vertical capillary tube of known geometry, and the solution flows under the effect of gravity. The measured quantity is the elution time needed for the free head of the fluid to cross a certain portion of the tube. The elution time is proportional to the viscosity of the fluid. If *t* is the elution time of the solution and $t_{0}$ is that of the solvent, the specific viscosity is obtained as:(2)$$\eta_{sp}=\frac{\eta_{sol}-\eta_{s}}{\eta_{s}}=\frac{t-t_{0}}{t_{0}}$$
where $\eta_{sol}$ and $\eta_{s}$ are the solution and solvent viscosities, respectively. The intrinsic viscosity is obtained as the zero-concentration limit of the $\eta_{sp}/C$ vs *C* function. Concerning the falling-ball viscometer, the viscosity is obtained by measuring the velocity of a sphere falling in the solution. The velocity of the sphere and the viscosity of the solution are related by Stokes’ law [67]. Once the viscosities of the solvent and the solutions are known, the intrinsic viscosity is extrapolated in the same way as for the Ubbelohde capillary.

Intrinsic viscosity measurements represent an easy, relatively fast, and cost-effective way to obtain information about the average molecular weight of PET. The drawback is that it is difficult to find suitable non-toxic solvents to prepare the dilute solutions.

### 4.2. Rotational Rheometry

Measurements on rotational rheometers have the advantage of ease of implementation and the possibility of reaching an ideally infinite number of strain units. Furthermore, dynamic shear oscillatory measurements allow the relaxation spectra of the samples to be obtained, which can be linked to changes in the molecular architecture. On the other hand, edge instabilities limit viscosity measurements at high shear rates in rotational devices. Conversely, capillary rheometry permits the measurement of the flow curves in a large shear rate range, but only information about steady-state viscosity can be obtained. Therefore, the two techniques must be regarded as complementary [68]. In terms of cost, capillary rheometers are generally less expensive than rotational rheometers. However, the latter are more versatile and easier to operate and allow for deep rheological characterization in both linear and nonlinear regimes. For this reason, they are preferred to capillary rheometers. However, if, for processing reasons, information about viscosity at a high shear rate is needed, this can only be achieved with capillary experiments. In all cases, the duration of the testing is proportional to the time scales to be probed.

In rotational rheometry, the sample is placed between a cone and a plate or parallel plates. The latter is the preferred geometry for linear viscoelastic measurements on polymer melts. In stress-controlled mode, the bottom plate is still, whereas the upper plate rotates under the action of a well-defined torque (*M*) imparted by a motor. The angular displacement (θ) is measured by an optical encoder integral with the motor. In strain-controlled mode, the bottom plate is attached to a motor that imparts a well-defined angular motion, whereas the upper plate is connected to a transducer that measures the resulting torque.

In both cases, the shear stress (σ), the shear strain (γ), and the shear rate ($\dot{\gamma }$) are related to the torque (*M*), angular displacement (θ), and angular velocity (Ω) by the following respective relationships [67]:(3)$$\begin{array}{ccc} \sigma & = & \beta \frac{2M}{\pi R^{3}} \end{array}$$(4)$$\begin{array}{ccc} \gamma & = & \frac{\theta r}{H} \end{array}$$(5)$$\begin{array}{ccc} \dot{\gamma } & = & \frac{\Omega r}{H} \end{array}$$
where *H* is the gap between plates, *R* is the plate radius and β is a correction factor smaller than unity that accounts for non-Newtonian behavior.

#### 4.2.1. Flow Curves

In a flow curve experiment, the material is subjected to a shear rate step, whereas the transient shear stress and viscosity are measured as functions of time. If the rate is low enough, then linearity is maintained throughout the test. The stress (or viscosity) undergoes a rapid increase before approaching a steady-state value in a time of order or the longest characteristic relaxation time of the material. At higher rates, the response becomes nonlinear. As PET is a shear-thinning material, the transient shear stress growth coefficient ($\eta^{+}\left(t\right)$) is expected to go reach a maximum before attaining a steady state. The flow curve, that is the function $\eta_{steady}=\eta_{steady}\left(\dot{\gamma }\right)$, is obtained by running a sequence of step-rate tests and plotting the steady-state viscosity as a function of the rate.

#### 4.2.2. Dynamic Tests

In dynamic oscillatory measurements, a sinusoidal strain wave (or stress wave, depending on whether the rheometer is strain- or stress-controlled) is applied, and the resulting stress (or strain) is measured. The strain wave is mathematically expressed as follows:(6)$$\gamma \left(t\right)=\gamma_{0}\sin\left(\omega t\right)$$
where $\gamma_{0}$ is the strain amplitude, and ω the angular frequency. In a linear regime, i.e., for low $\gamma_{0}$ values, the resulting stress is sinusoidal, with the same frequency and a phase shift (δ) that depends on the viscoelasticity of the material:(7)$$\sigma \left(t\right)=\sigma_{0}\sin\left(\omega t+\delta \right)$$

By normalizing the stress relative to the strain amplitude ($\gamma_{0}$), the output can be written in terms of a storage (or elastic) modulus ($G^{\prime }$) and a loss (or viscous) modulus ($G^{\prime \prime }$):(8)$$\frac{\sigma (t)}{\gamma_{0}}=G^{\prime }\sin\left(\omega t\right)+G^{\prime \prime }\cos\left(\omega t\right)$$
where
(9)$$G^{\prime }=\frac{\sigma_{0}}{\gamma_{0}}\cos\left(\delta \right)$$
and
(10)$$G^{\prime \prime }=\frac{\sigma_{0}}{\gamma_{0}}\sin\left(\delta \right)$$

The ratio $G^{\prime \prime }/G^{\prime }=\tan\left(\delta \right)$ is referred to as a loss factor, as it indicates the attitude of the material to dissipate energy rather than store it. Other viscoelastic quantities frequently used in the description of dynamic measurements are the complex modulus ($|G^{*}|$)
(11)$$|G^{*}|=\frac{\sigma_{0}}{\gamma_{0}}=\sqrt{{\left(G^{\prime }\right)}^{2}+{\left(G^{\prime \prime }\right)}^{2}}$$
and the complex viscosity ($|\eta^{*}\left|=\right|G^{*}|/\omega$). Oscillatory tests are used in different ways to test material properties. If one wants to evaluate the stability at a fixed temperature, then a single oscillation at a fixed frequency and strain amplitude is set, and the viscoelastic moduli are measured as functions of time (time sweep test). To assess the linear regime, a sequence of oscillations at a fixed frequency and increasing strain is performed (strain sweep test). In a linear regime, the viscoelastic moduli are constant relative to deformation, whereas they become functions of deformation when the linear regime is exceeded. To test the viscoelastic properties at different time scales, a series of oscillations at fixed strain amplitude and varying frequency are carried out (frequency sweep test).

Another possibility is to build a periodic strain wave that contains multiple harmonics and decompose the resulting stress wave to obtain the response at different frequencies (multiwave spectroscopy).

#### 4.2.3. Uniaxial Extensional Tests

For a long time, extensional melt rheology measurements have been strongly limited by intrinsic experimental difficulties. Today, uniaxial extensional tests can be reliably and affordably performed on rotational rheometers equipped with ad hoc devices. This possibility was pioneered by Sentmanat [69], who designed the so-called Sentmanat extensional rheometer (SER), which is made of two counter-rotating cylinders whose motion is paired by gears. The sample consists of a rectangular bar placed on the two rotating cylinders. By imposing an angular velocity on one of the cylinders, they rotate and stretch the sample bar. Several variations of his original design are now available for most commercial rheometers.

As the sample bar is placed horizontally on the cylinders, the test temperature must be high enough to ensure adhesion on the cylinders but low enough to prevent sagging.

The Hencky strain rate (${\dot{\varepsilon }}_{H}$) applied to the sample can be expressed as follows:(12)$${\dot{\varepsilon }}_{H}=\frac{2\Omega R}{L_{0}}$$
where $L_{0}$ is the centerline distance between the two drums.

The stretch force is related to the torque by the following expression:(13)M=2FR

In uniaxial extension, the sample cross section (A(t)) decreases exponentially over time with respect to the initial value ($A_{0}$) according to the following equation:(14)$$A\left(t\right)=A_{0}\exp\left(-{\dot{\varepsilon }}_{H}t\right)$$

At a constant strain rate, the extensional stress growth coefficient ($\eta_{E}^{+}\left(t\right)$) is evaluated as follows:(15)$$\eta_{E}^{+}\left(t\right)=\frac{F(t)}{{\dot{\varepsilon }}_{H}A\left(t\right)}$$

In a uniaxial extension experiment, ${\dot{\varepsilon }}_{H}$ is imposed by imposing the angular velocity (Ω), and the stretching force *F* is measured through a torque reading. Then, knowing the initial cross section ($A_{0}$), the quantity ($\eta_{E}^{+}\left(t\right)$) is readily evaluated.

### 4.3. Capillary Rheometry

Capillary rheometry is used to measure the viscosity of the fluid at high shear rates. In a capillary rheometer, the test fluid contained in a reservoir is forced to flow through a small-diameter tube connected to the reservoir. There are two ways to perform a capillary experiment: by imposing either the flow rate or the pressure at the entrance of the capillary. In the first case, the fluid is pushed through the capillary by a piston moving at a given velocity, and the pressure at the entrance of the capillary is measured by a pressure transducer. In the second case, the fluid is driven by a compressed gas at a known pressure, and the flow rate is measured by a scale or by optical methods [70]. The shear stress ($\tau_{w}$) at the wall of the capillary is related to the pressure drop (ΔP) at the capillary ends:(16)$$\tau_{w}=\frac{R}{2}\frac{\Delta P}{L}$$

If the fluid is Newtonian, the Hagen–Poiseuille law holds:(17)$$\Delta P=\frac{8}{\pi }\frac{L}{R^{4}}\eta Q$$
where *L* is the capillary length, and *Q* is the volumetric flow rate. Hence, the apparent shear rate (${\dot{\gamma }}_{app}$) is equal to the actual shear rate ($\dot{\gamma }$):(18)$${\dot{\gamma }}_{app}=\frac{4Q}{\pi R^{3}}$$
and the viscosity is simply obtained as $\eta =\tau_{w}/\dot{\gamma }$. If the fluid is non-Newtonian, then a correction must be performed:(19)$${\dot{\gamma }}_{w}={\dot{\gamma }}_{app}\left(\frac{3}{4}+\frac{1}{4}\frac{d\ln Q}{d\ln\tau_{w}}\right)$$
and the actual viscosity is obtained as $\eta =\tau_{w}/{\dot{\gamma }}_{w}$. Note that to calculate the shear rate from Equation (Equation 19), the derivative of the flow rate as a function of the wall shear stress must be evaluated; hence, multiple measurements must be run. For power-law fluids, such a derivative is equal to the exponent of the power law. Furthermore, as the transducer is generally placed inside the reservoir, the measured shear stress includes the entrance pressure drop. To eliminate such a contribution, an additional correction is performed by running measurements with different capillary lengths [67].

**Table 3 materials-16-03358-t003:** Summary of relevant rheological tests on PET materials in the literature.

Type	Purpose	Reference
Ubbelohde/falling-ball viscometry	Determination of the $M_{w}$	[51]
Time sweep test	Effect of thermo-oxidative degradation on the material and evaluation of the chain extender effectiveness on linear PET	[40,57,66,71,72]
Frequency sweep test	Determination of viscoelastic properties of the system	[33,46,48,51,61]
Non-linear extension test	Examination of elongation properties examination and detection of long-chain branches in the system	[39,42,53,55,62]
Capillary test	Flow curve analysis	[44,52,73,74,75]

## 5. Stability of PET

Polyethylene terephthalate can easily degrade under processing conditions. The most relevant factors promoting degradation of PET are moisture, oxygen, heat and UV light [76,77,78]. Concerning the latter, photodegradation of PET can proceed either by direct cleavage of the ester bond or via radical reactions [79]. Radicals, in turn, can also promote crosslinking [80,81], leading to an increase in the viscoelastic properties.

There are contrasting reports in literature about the impact of degradation on the viscoelastic properties of PET. Early works by Marshall and Todd [82] and Buxbaum [83] evidenced that the random chain scission process caused by thermal degradation leads to a decrease in molecular weight, affecting both melt and intrinsic viscosities. A decrease in the viscoelastic properties was reported to occur in both air and nitrogen atmospheres [41,46,84,85,86]. However, Guenther and Baird [85] observed that the degradation of PET was smaller in a nitrogen atmosphere compared to air. In their work, Seo and Cloyd showed that the degradation rate of PET can be expressed as the sum of two contributions, the first of which is due to hydrolysis, which occurs due to residual moisture in the sample, and the second of which is due to thermo-oxidative degradation [87]. Both contributions lead to the cleavage of PET linear chains, which results in a decrease in molecular weight and, thus, in viscosity.

Conversely, more recent works reported an increase in melt viscosity of PET when measured in a nitrogen environment [44,47,72,88,89]. The increase in melt viscosity in an inert atmosphere was attributed to crosslinking due to small amounts of oxygen [88,90]. The disagreement on the degradation of PET under nitrogen is probably related the different type of catalyst used to synthesize the polymer.

It is worth noting that thermal degradation also occurs during chain extension, resulting in a nonmonotonic trend of relevant quantities such as elastic modulus [72] or torque [91] over time. For example, the viscosity increases at the beginning of the process, when the chain extension rate is faster than chain scission. As chain extension advances further, the extension rate becomes smaller than the scission rate, resulting in a decrease in viscosity.

The effects of PET degradation were investigated through different rheological tests. A first method involves the analysis of torque as a function of time; usually, this type of test is carried out inside a mixer or an extruder, and trends can also be tracked during sample preparation [33]. Andrade et al. investigated the evolution of torque over time for PET samples processed inside a mixer at various temperatures and rotation speeds. Based on their experiments, they concluded that PET resins always undergo degradation, even if minimal, despite different experimental conditions [92].

Another efficient method to evaluate PET stability as a function of time and under different experimental conditions is the performance of time sweep tests.

In 2015, Härth et al. [71] examined the stability of a commercial PET resin via time sweep tests by studying the influence of process variables on polymer degradation. As an example of a rheological test to monitor degradation, Figure 3 reports the viscoelastic moduli as functions of time; changes in the $G^{\prime }$ and $G^{\prime \prime }$ curves can be evaluated depending on the inert gas used, the type of atmosphere (air or nitrogen), and the water content of the sample in order to assess the influence of drying.

Figure 3 shows the decreasing trends of viscosity due to thermo-oxidative degradation when tests were carried out in an air atmosphere; on the other hand, when the samples were exposed to nitrogen, viscosity increased over time because of the predominance of polycondensation reactions among the linear PET chains, in contrast to the contribution of degradation. In addition, the effectiveness of drying can be appreciated by noticing that viscosity values are about a decade lower for samples with a higher water content; in this case, hydrolysis prevails over other reactions. Another way to study the degradation of PET is via time-resolved rheometry [73]. Such a method provides more information with respect to a time sweep test, as it measures the evolution of the frequency spectrum over time.

## 6. Torque Measurements during Reactive Extrusion

An indirect measurement of the increase in the viscoelastic properties during the chain extension process can be obtained by torque measurements during batch mixing. To this end, the sample and the chain extender are loaded in a batch mixer at a fixed temperature, and the torque needed to keep the system under constant stirring is measured as a function of time. As the chain extension process advances, the viscoelasticity of the system increases; hence the torque needed to maintain the stirring also increases. Note that during mixing, the flow is non-viscometric; hence, it is not possible to relate the measured torque to specific viscoelastic quantities such as viscosity or viscoelastic moduli. In addition, as chain extension, branching and degradation occur simultaneously, it is not possible to discern the effect of each of these processes on the torque increase. However, several authors have used torque measurements to follow the chain extension process [18,33,41,46,51,58,91,92,93,94,95]. Generally, it is observed that the addition of a chain extender to virgin PET causes an increase in the torque over time (see Figure 4). In particular, the higher the concentration of the chain extender, the greater the relative increase in torque with respect to virgin PET. After the chain extension process is completed, if further mixing is carried out, a decrease in torque due to degradation is observed [58,61]. By analyzing torque vs. time trends, Arayesh et al. [33] evaluated the effectiveness of four different chain extenders in the production of branched PET by reactive mixing; they observed an increase in torque for the processed samples, suggesting an improvement in system viscosity due to branching. Xanthos et al. made the same observation in 2004 [18]; in their work, they measured the increase in torque vs. time for PET when processed with different concentrations of TGIC. In general, testing the efficacy of a system modifier (i.e., branching agents, functionalizers, etc.) by examining torque as a function of time is a common practice; this technique was also employed by Avarzman [93], Lacoste [94], Xiao [58], and Incarnato [46]. Instead, Liu et al. [51] evaluated the competition between the three reactions of degradation, branching, and crosslinking by comparing the torque profiles for both unprocessed and modified PET at various PMDA contents. The effect of thermal degradation in terms of torque was also measured by Dhavalikar et al., who examined PET both with and without a chain extender at two different temperatures [41].

## 7. Linear Rheology

### 7.1. Dynamic-Frequency Sweep Measurements

As reported in Table 1, the melting temperature of PET is approximately 250–255 °C; hence, rheological measurements are generally performed between 260 and 290–300 °C. Above the latter temperature, strong degradation occurs within a short duration, limiting the feasibility of the tests. In the temperature range of 250–300 °C, PET behaves as a thermorheologically simple material, that is, by performing rheological tests at different temperatures, a master curve can be built. However, in this range, the material is well above the glass transition temperature; hence, the dependence of the horizontal shift factor upon temperature is Arrhenius-like (see Equation (Equation 20)). For this reason, the extension of the frequency range is generally limited.

Virgin PET has a linear architecture with high polydispersity and an average $M_{w}$ lower than $10^{5}$ g/mol. Since the entangled molecular weight is between 1450 and 2150 g/mol, virgin PET is generally a well-entangled material. Nonetheless, since rheological measurements are performed far from the glass transition, dynamic measurements generally exhibit a large portion of terminal behavior. In this regime, the viscous modulus shows a constant slope of 1 and is larger than the elastic modulus, which shows a constant slope of 2. The elastic modulus sometimes shows a low-frequency plateau [33]. However, this must be regarded as an experimental artifact due to the low torque limits of the instrument or incipient degradation, rather than a material property [96]. Analysis of the horizontal shift factors ($a_{T}$) allows conclusions to be drawn with respect to branching based on the flow activation energy ($E_{a}$). The latter can be obtained by fitting $a_{T}$ using the following Arrhenius equation:(20)$$log\left(a_{T}\right)=\frac{E_{a}}{R}\left(\frac{1}{T}-\frac{1}{T_{\mathrm{ref}}}\right)$$
where *R* is the gas constant, and $T_{\mathrm{ref}}$ is the reference temperature. The same trend is observed if the zero-shear viscosity ($\eta_{0}$) is plotted instead of $a_{T}$. A strong linear increase in the flow activation energy is a signature of LCB [42,55].

If the results are presented in terms of complex viscosity, a large plateau of $\eta^{*}$ is observed. Weak shear thinning occurs only at high frequencies. When PET is reacted with a chain extender, the average molecular weight increases; hence, the longest relaxation time becomes larger (Figure 5).

If the chain extender is multifunctional, the molecular architecture changes from linear to a branched. Branching does not alter the value of the entanglement plateau modulus. However, the hierarchical relaxation modes of the branched architecture result in power-law behavior before the terminal regime is approached. In terms of complex viscosity, when chain extension occurs, the plateau value of the zero-shear viscosity increases, and shear thinning starts from lower frequencies. The higher the degree of branching, the broader the transition from the Newtonian behavior to the thinning region.

Irrespective of the particular chain extender used, the increase in the molecular weight and the degree of branching are directly proportional to the concentration of the chain extender. Subsequently, the zero-shear viscosity increases with increasing chain extender concentration. At high frequencies, the shear-thinning parts of the different master curves ideally collapse. When the degree of branching is increased, some crosslinking cannot be excluded. The samples of C-PET have a non-negligible gel content. The latter can be determined by dissolving the samples and weighing the undissolved portion. From a rheological standpoint, when a high degree of branching and partial crosslinking occur, the frequency response shows features of near-critical gels. The viscoelastic moduli are parallel and show power law behavior with respect to frequency [98].

As shown in the next sections, there are other possibilities to present linear rheological data in order to better highlight the link between the degree of branching and the viscoelastic measurements.

#### 7.1.1. Van Gurp–Palmen Plots

One possibility is to use Van Gurp Palmen (VGP) plots, that is, to report the phase angle as a function of the complex modulus [99]. An important feature of this plot is that it reveals whether the material is thermorheologically simple or not without applying the time–temperature superposition principle. If thermorheological simplicity is verified, the data at different temperatures collapse on a single master curve.

The shape of VGP plots can provide information on the increase in polydispersity and possible branching phenomena [100].

Van Gurp–Palmen plots were used to detect LCB in PET reacted with different chain extenders [42,53,101]. With respect to the virgin sample, a flattening of the curve and a shoulder appeared in samples with chain extenders. The shoulder was attributed to LCB. When branching occurs to a large degree, the shoulder becomes more pronounced, and a minimum is observed in the VGP curves. Figure 6 reports the VGP plot for virgin and reacted PET samples from [53]. LCB PET was produced by reacting virgin L-PET samples with PMDA, whereas C-PET was produced by reacting virgin L-PET with Joncryl. C-PET has a much higher degree of branching; hence, a minimum appears in the VGP plot. Thanks to VGP plots, Kruse [42] showed the different effects of two chain extenders, PMDA and TGDDM, on linear PET chains. He succeeded in demonstrating the development of two different branched molecular architectures by comparing the phase-angle shifts for the two systems. Benvenuta Tapia [101] evaluated the occurrence of LCB in recycled PET due to the addition of reactive copolymers to the system.

#### 7.1.2. Cole–Cole Plots

A Cole–Cole plot represents the viscous modulus as a function of the elastic modulus. For a purely Maxwell material, the Cole-/Cole plot is a semicircle; hence, the low-$G^{\prime }$ part of the curve has a positive slope and corresponds to long time scales, whereas the high-$G^{\prime }$ part of the plot has a negative slope and is associated with short time scales. As mentioned above, it is difficult to measure the elastic plateau region of PET in frequency sweep experiments because of the relatively low molecular weight and short relaxation time [66]. For this reason, many studies have focused on the low-$G^{\prime }$ part of the curve [40,47,48,52,61,66,102,103]. Several authors used Cole–Cole plots to estimate the effects of various chain extenders on linear PET systems. Daver [47] evaluated the efficiency of PMDA; Wang [61] studied a step-chain extension process with two different additives, SAG-008 and TGDDM; Dolatshah [48] examined the effect of a catalyst, DBTDL, on the chain-extension reaction; and Liu [40] analyzed the combination of two different chain extenders, PMDA and TADE. Cole–Cole plots were reported to be sensitive to polydispersity and the degree of branching but independent of molecular weight [104]. The line $G^{\prime }=G^{\prime \prime }$ sets the boundary between elastic and viscous behavior, as above this line, the loss modulus is larger than the elastic modulus. In general, virgin PET samples are above the line $G^{\prime }=G^{\prime \prime }$. The main outcome of Cole–Cole analysis in the literature on PET is that a higher degree of branching shifts the curve towards higher $G^{\prime }$ values with constant $G^{\prime \prime }$ values [48,58,61], as shown in Figure 7. It can also be noted that a higher degree of branching induces a shoulder in the plot at high $G^{\prime }$ values. In some cases, a high branching degree induces a minimum in this region [48].

In addition to investigating the degree of branching, Cole–Cole plots can be used to deduce other important aspects. Souza et al. [103] evaluated the elastic response of commercial PET as the water content in the system changed; Cruz et al. [66], on the other hand, quantified the degradation induced by reprocessing contaminants and SSP on recycled PET.

### 7.2. Intrinsic Viscosity Measurements

Intrinsic viscosity measurements are useful to evaluate the molecular weight of PET, as well as to detect LCB.

As far as the chain-extension process is concerned, intrinsic viscosity measurements are used to obtain the molecular weight of the reacted and unreacted PET. An increase in the chain extender concentration during the preparation of branched PET leads to an increase in the intrinsic viscosity of the samples. In particular, the higher the concentration of the chain extender, the higher the intrinsic viscosity of the sample [61,105].

## 8. Nonlinear Rheological Properties

### 8.1. Flow Curves

Flow curves can be obtained either by rotational rheometry, capillary devices, or a combination of both techniques. As mentioned above, the first option does not allow very large values of the shear rate ($\dot{\gamma }$) to be obtained. With rotational rheometry, the maximum attained shear rate is of the order of 100 $s^{-1}$, whereas capillary rheometers allow for the measurement of steady-viscosity up to $10^{4}$–$10^{5}$$s^{-1}$. Figure 8 shows the flow curves of two PET samples from reference [19]—one modified with PMDA (PET4) and an unmodified one (PET8). The unmodified linear PET shows a Newtonian plateau before the thinning region, whereas branching induces a larger zero-shear viscosity, which could not be measured in the experimental range. At high rates, the two samples undergo shear thinning behavior. Note that as branching and polydispersity strongly increase during the chain-extension process, the thinning slope of linear and branched polymers is not necessarily the same. The data can be fitted with empirical models such as the Cross equation—$\eta /\eta_{0}={\left[1+{\left(\tau \dot{\gamma }\right)}^{1-n}\right]}^{-1}$, where $\eta_{0}$, τ and *n* are fitting parameters. The fit of both curves shown in Figure 8 yields thinning exponents of 0.71 for PET8 and 0.51 for PET4.

Experiments conducted at high rates in capillary rheometers are faster and less prone to degradation. In reference [106], it was shown that a discrepancy occurs between flow curves obtained with rotational rheometers and those obtained with capillary rheometry due to degradation.

### 8.2. Extensional Rheology

Extensional viscosity is highly sensitive to the degree of branching. In particular, a high branching degree induces marked strain hardening with respect to the linear counterpart. For this reason, uniaxial extension is largely used to study the structural modification of PET during reactive extrusion. Figure 9 shows the tensile growth function of PET modified with two different chain extenders.

Strain hardening, that is, the sudden and considerable increase in the transient elongation viscosity at high stretching rates, can be measured by defining the so-called strain-hardening factor
(21)$$X=\frac{\eta_{E}^{+}\left(t,\dot{\varepsilon }\right)}{\lim_{\dot{\varepsilon }\to 0}\eta_{E}^{+}\left(t,\dot{\varepsilon }\right)}$$

The Hencky strain is simply obtained as $\varepsilon_{H}=\dot{\varepsilon }t$. Hardening increases with increasing concentration of the chain extender, as shown in Figure 10.

## 9. Molecular Rheological Models

Many efforts have been made to model the viscoelastic properties of both virgin and reacted PET, with a dual objective. On the one hand, constitutive relations properly describing the rheological behavior of PET are relevant for processing and simulations. On the other hand, molecular models help to link the viscoelastic properties to the molecular structure in order to quantify the effect of chain extenders.

On the continuum mechanics side, a multimode Phan-Thien constitutive relation was demonstrated to successfully predict the behavior of PET resins in both shear and extensional flows [106]. The continuum approach is useful to implement fluid dynamics simulation of the process. On the other hand, it does not provide information about the modification of the molecular structure during chain extension. Molecular models are much more interesting, as they can directly relate rheology to the molecular architecture. Models describing branched structures have been used to analyze the viscoelasticity of PET materials, mainly to quantify the degree of branching induced by reactive extrusion. As chain extension is a random branching process, models such as the branch-on-branch (BOB) algorythm [108] are the most suitable for inferring the polymer structure from the viscoelastic properties. The BOB model was successfully used to fit the linear viscoelasticity of branched PET and to extract information about the degree of branching in LCB-PET (Figure 11 from reference [40]). Such a model can predict the linear and nonlinear viscoelasticity of a regular or randomly branched structure in both linear and nonlinear regimes and in both shear and uniaxial extension [109].

Implementing the BoB model using rheological data allows for quantification of the degree of branching of the final melts, providing information on the efficiency of the chain-extension process. However, inferring the molecular structure distribution from rheological measurements is an ill-posed problem, as different combinations of molecular structures can lead to the same viscoelastic response. Therefore, in the absence of other information about the distribution of molecular structures, for example, from the synthesis, the results of the modeling must be regarded as purely indicative. Another model used to predict the degree of LCB of PET according to uniaxial extensional behavior is the molecular stress function (MSF) model proposed by Wagner [110], which is based on the tube model proposed by Doi and Edwards [111]. For LCB samples, a free parameter (β) is introduced in the evolution equation of the stress function (*f*). The β parameter is related to the number-averaged molecular weight ($M_{n}$) of the branches in the molecule. The MSF model was successfully implemented to predict the uniaxial extensional behavior of PET reacted with PMDA and TGDDM [42]. The strength of this approach lies in its ability to easily quantify the average degree of molecular branching through the β parameter. For example, it was demonstrated that β correlates with the concentration of the chain extender [42]. However, in this case, it is also difficult to draw definitive conclusions about the distribution of molecular structures without any additional information from other analytical techniques.

## 10. Conclusions

The literature examined in this review highlights the importance of rheology as a tool to analyze the variation of molecular architecture and properties of PET during the reactive extrusion process. Rheological measurements provide information on different aspects of the process. Time-resolved tests allow for measurement of the stability of PET at different temperatures in different environments. Furthermore, they allow for monitoring of increases in the viscoelastic properties. The relaxation spectrum of the final product can be derived from frequency sweep tests, based on which information on the final microstructure can be inferred. Information on the extensional properties are directly provided by non-linear extensional tests, which are fundamental for the foaming process. In conclusion, rheology allows for the establishment of a link between the process variables, i.e., concentration of the chain extender, and the final product, i.e.,the degree of branching, allowing for control of the output of the chain extension based on the process parameters.

In spite of the considerable efforts made to date, the knowledge of the chain extension process remains semiempirical. A predictive tool to extrapolate the change in molecular weight distribution and architecture occurring after the extrusion process is currently lacking.

## 11. Perspectives

To achieve fine control of the extrusion process, future research efforts should be concentrated on evaluating changes in molecular weight distribution and architecture occurring during chain extension. Such a task requires a combination of knowledge provided by different analytical methods involving synthesis, experimental rheology and modeling.

Prediction of molecular weight distribution based on linear viscoelastic properties has attracted substantial interest from rheologists for the last 40 years [112,113,114], although literature studies have focused polymer materials other than PET. Most literature on the topic concerns linear polymers, as they are easier to model compared to branched structures. Modeling approaches are based on the so-called mixing rule [115]. The linear relaxation modulus (or the viscosity) is expressed as a convolution integral involving the molecular weight distribution function and the relaxation function of the monodisperse linear polymer [112]. Such an approach enables determination of the rheological response from the MWD and vice versa. However, the latter is an ill-posed problem with multiple solutions.

Starting from the reagents, one can use statistical approaches to predict the final MWD and chain architectural distribution. Then, by using a mixing rule, one can obtain the rheological properties. With such a chain of knowledge, the final viscoelastic properties of the mixture can be tuned starting from the synthetic process, for example, to achieve specific elongation properties for foaming. For this reason, developing such a predictive tool would be highly desirable. Nonetheless, few attempts have been made to link synthesis and rheology for simple branched structures [116]. At the moment, a major objective is to refine statistical approaches for the prediction of the synthetic output of chain extension. Furthermore, it is also challenging to postulate a suitable mixing rule for branched architectures because long chain branching and polydispersity can have the same influence on rheological properties [117].

## Figures and Tables

**Figure 1 materials-16-03358-f001:**
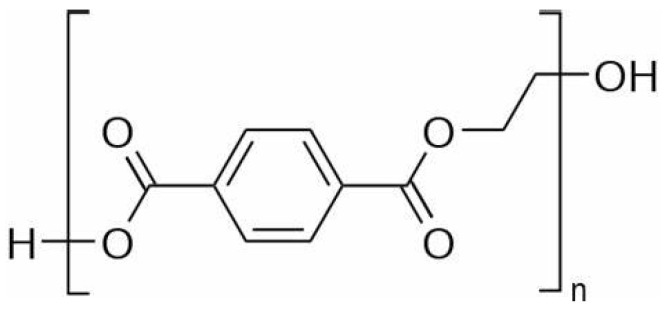
Repeating unit of PET.

**Figure 2 materials-16-03358-f002:**
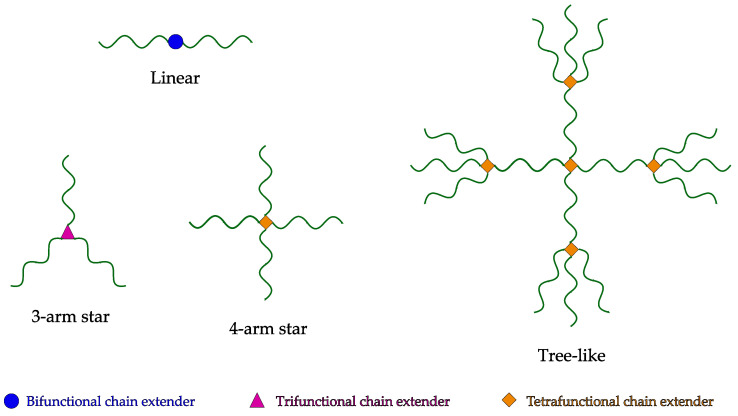
Different structures arising from the chain extension process.

**Figure 3 materials-16-03358-f003:**
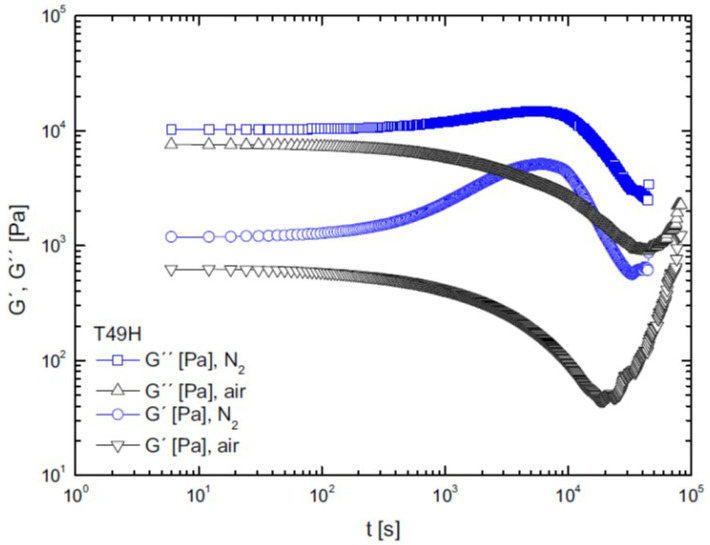
Viscoelastic moduli vs. time [72].

**Figure 4 materials-16-03358-f004:**
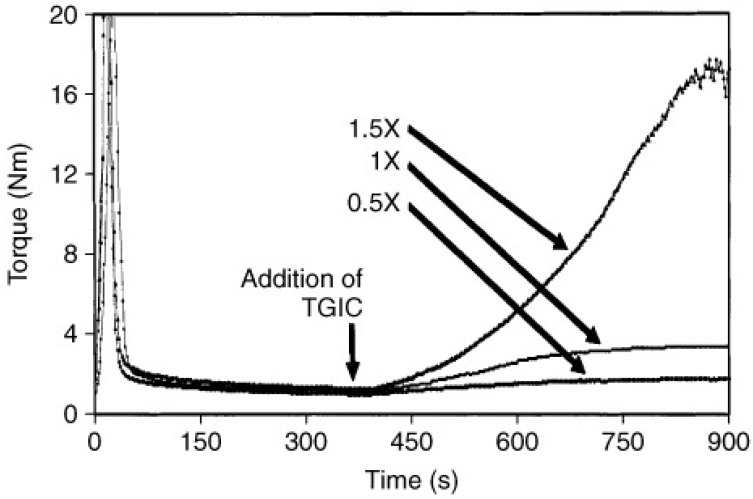
Typical torque versus time data of PET modification in a batch mixer with TGIC added at different concentrations [18].

**Figure 5 materials-16-03358-f005:**
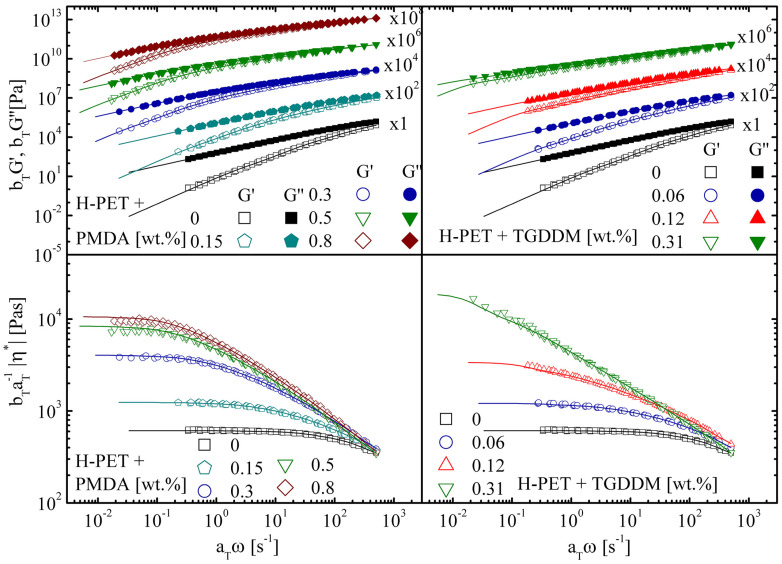
Time–temperature superposition of the storage and loss modulus and complex viscosity (symbols) for H-PET processed with different concentrations of PMDA and TGDDM. Fit according to the generalized Maxwell model (solid lines). Reference temperature: 265 °C; reprinted from [97].

**Figure 6 materials-16-03358-f006:**
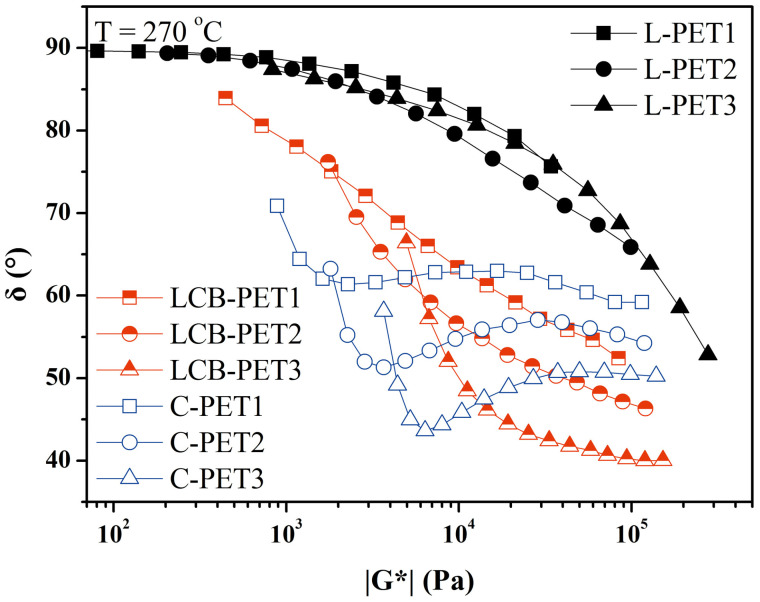
Van Gurp–Palmen plots of linear and LCB PET [53].

**Figure 7 materials-16-03358-f007:**
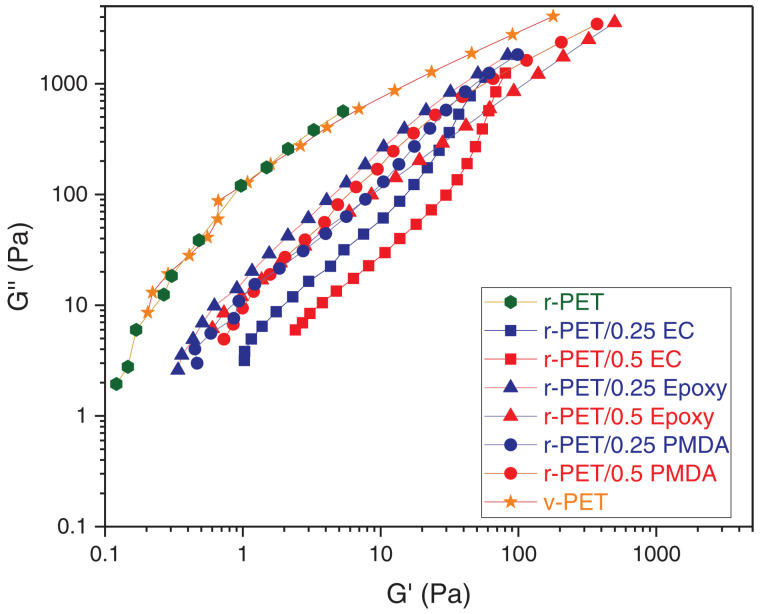
Cole–Cole plot for various chain-extended poly(ethylene terephthalates) (PETs), where r-PET is recycled PET, and v-PET is virgin PET, and 0.25 and 0.5 are the weight percent of chain extenders (reprinted from [52]).

**Figure 8 materials-16-03358-f008:**
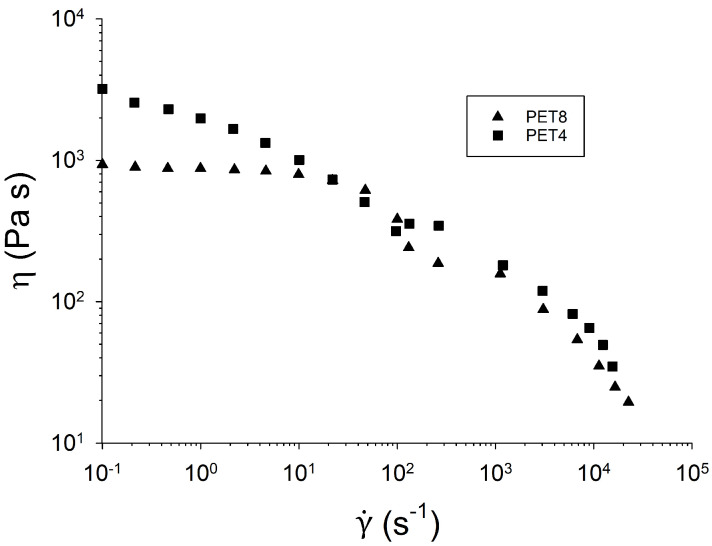
Melt viscosity comparison of unmodified PET8 and modified PET4 at 290 °C. Reprinted from [19].

**Figure 9 materials-16-03358-f009:**
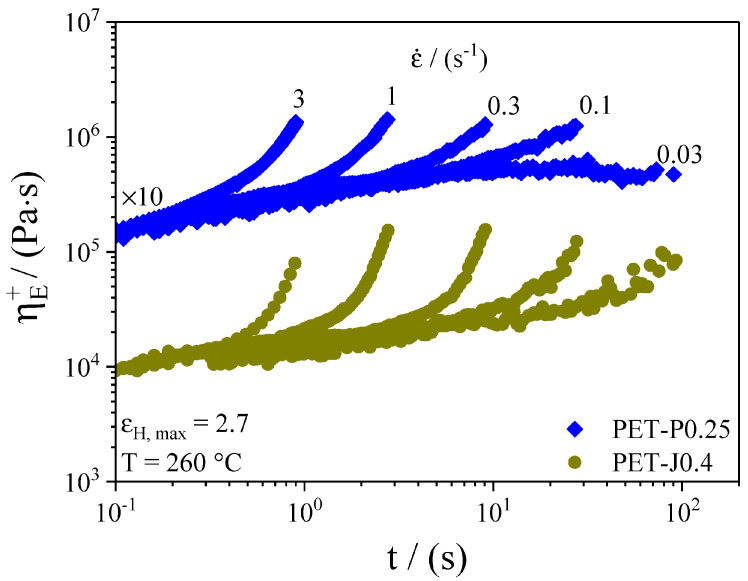
Time-dependent elongation viscosity at different elongation rates for PET-J0.4 and PET-P0.25. The PET-P0.25 curves are shifted by 10, as indicated in the figure. Reprinted from [50].

**Figure 10 materials-16-03358-f010:**
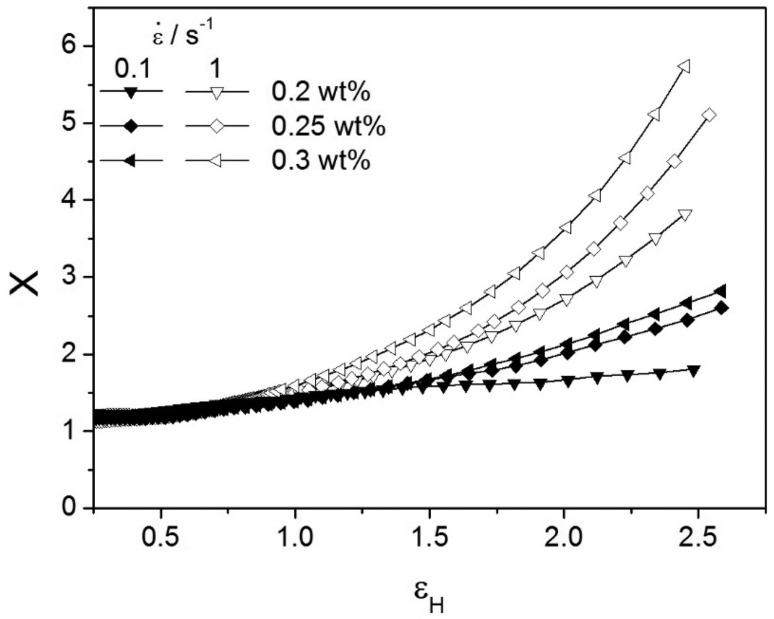
Strain-hardening factor as a function of Hencky strain for two strain rates and for 0.2, 0.25, and 0.3 wt % PMDA samples. Reprinted from reference [107].

**Figure 11 materials-16-03358-f011:**
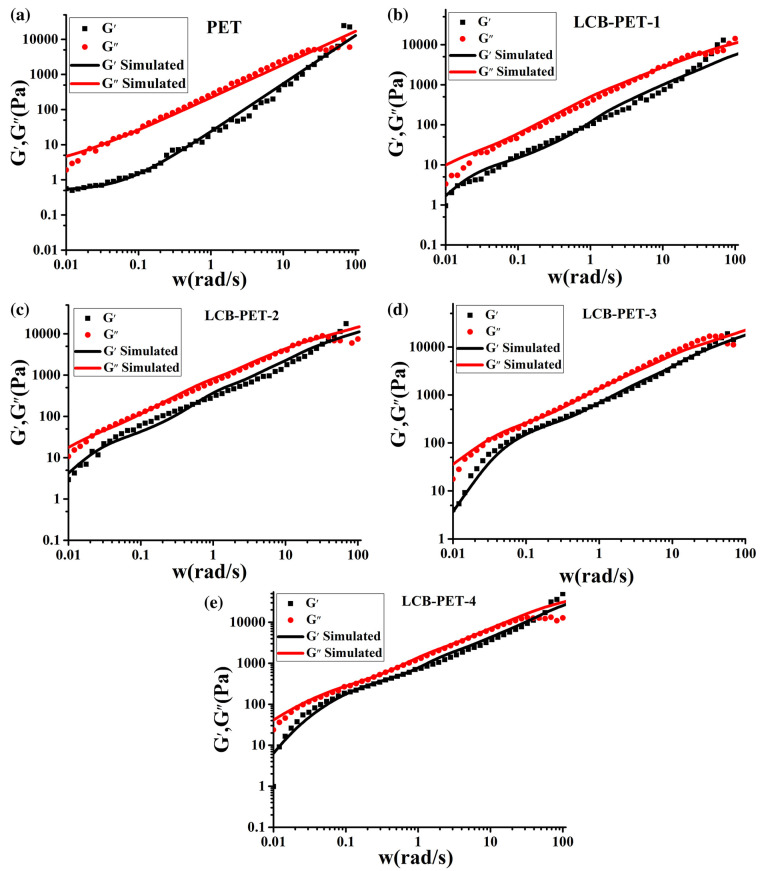
Linear viscoelastic spectra at 260 °C for (**a**) PET; (**b**) LCB-PET-1; (**c**) LCB-PET-2; (**d**) LCB-PET-3; and (**e**) LCB-PET-4. From Liu et al. [40].

**Table 1 materials-16-03358-t001:** Relevant chemical and physical properties of PET.

Property	Value	Reference
Density	1.33–1.45 g/cm${}^{3}$ ^a^	[20,21,22,23]
Glass transition temperature	69–85.4 °C	[24,25,26]
Melting temperature	250–255 °C	[24,27,28]
Degradation onset temperature	380–420 °C	[29,30]
Plateau modulus, $G_{N}^{0}$	3.5 MPa ^b^	
Monomeric molecular weight, $M_{0}$	192.2 g/mol	
Flory characteristic ratio, $C_{\infty }$	4.20–5.83	[31]
Entanglement molecular weight, $M_{e}$	1450–2120 g/mol	[31,32]

^a^ Depending on the degree of crystallinity; ^b^ evaluated as $G_{N}^{0}=\left(\rho RT\right)/M_{e}$, with $M_{e}=1700$ g/mol and T=533.15 °C.

## Data Availability

Not applicable.

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
