# Peer review of "Rheology of Recycled PET"

_materials, 2023, doi:10.3390/ma16093358_

Round 1

Reviewer 1 Report

This manuscript briefly reviewed relevant rheological studies associated with the recycling of PET through the reactive extrusion process. The literature examined in this review highlights the importance of rheology as a tool to analyze the variation of molecular architecture and properties of PET during the reactive extrusion process. However, this paper is not suitable for publication in this form, which needs to be rewritten in light of the recommendations mentioned below.

Some major concerns are shown as following:

1. There must be Spaces between numbers and units.

2. The reference format must be uniform.

3. Introduction: Are there any review articles on Rheology of recycled PET? It is necessary to make clear in this section how this review differs from other reviews in the literature. Authors should include the novelty and importance of this review.

4. There are some polyester-related publications, and the author may cite them in the manuscript to enrich the introduction, for example: Adv. Fiber Mater., 2022, 4, 119; Adv. Fiber Mater., 2021, 3, 180.

4. The authors should summarize the advantages and disadvantages of different chain extenders and compare the advantages and disadvantages of different chain extenders.

5. In section 3, the authors claim to summarize the correlation between the chain extension process of different chain extenders and the changes in viscoelastic properties of PET. However, there is no relevant content in the article, please add relevant content in the revised manuscript.

6. In section 4, the authors must compare the advantages and disadvantages of different Rheological methods for testing PET materials (including but not limited to the scope of application of the testing method, ease of operation, accuracy of the test, test time and cost, etc.).

7. Will PET degrade when chain extender is used for chain extension processing? Is there any relevant literature on preventing PET degradation during chain extension processing? If so, the author should add the appropriate contents.

8. In section 9, the authors should comment on the scope, benefits, and limitations of the various models.

Reviewer 2 Report

The manuscript titled "RHEOLOGY OF RECYCLED PET" provides a brief review of the relevant rheological studies associated with the recycling of polyethylene terephthalate through the reactive extrusion process. The paper aims to highlight the importance of rheology as a tool to analyze the variation of molecular architecture and properties of PET during the reactive extrusion process. The conclusion section provides useful insights into the significance of rheology and the limitations of the existing knowledge on the chain extension process. However, there are a few issues that need to be addressed before the manuscript can be accepted for publication. The conclusion section should be separated from the Perspectives. The conclusions section provides a good summary of the importance of rheology in controlling the chain extension process of PET. However, it is important to note that a predictive tool to extrapolate the change in molecular weight distribution and architecture is currently missing. It would be beneficial to discuss potential avenues for future research to address this issue.

Overall, the manuscript has the potential to make a significant contribution to the field of recycled PET rheology. However, minor revisions are required to address the issues outlined above.

Round 2

Reviewer 1 Report

The authors have revised the manuscript carefully based on referees' comments. The scientific quality of this paper is improved. The manuscript can be accepted at the present version.